# Better Pandemic Influenza Preparedness through Adjuvant Technology Transfer: Challenges and Lessons Learned

**DOI:** 10.3390/vaccines9050461

**Published:** 2021-05-05

**Authors:** Céline H. Lemoine, Reviany V. Nidom, Roland Ventura, Setyarina Indrasari, Irine Normalina, Kuncoro Puguh Santoso, Francis Derouet, Christophe Barnier-Quer, Gerrit Borchard, Nicolas Collin, Chairul A. Nidom

**Affiliations:** 1Institute of Pharmaceutical Sciences of Western Switzerland, University of Geneva, Rue Michel-Servet 1, 1221 Geneva, Switzerland; gerrit.borchard@unige.ch; 2Vaccine Formulation Institute, Chemin des Aulx 14, 1228 Plan-les-Ouates, Switzerland; roland.ventura@vformulation.org (R.V.); nicolas.collin@vformulation.org (N.C.); 3Professor Nidom Foundation, Surabaya 60298, Indonesia; reviany@pnfinstitute.org (R.V.N.); setyarina_ire@pnfinstitute.org (S.I.); irine_normalina@pnfinstitute.org (I.N.); kuncoropuguhsantoso@pnfinstitute.org (K.P.S.); nidomca@fkh.unair.ac.id (C.A.N.); 4Faculty of Veterinary Medicine, Universitas Airlangga, Surabaya 60115, Indonesia; 5Centre Laboratoire d’Epalinges (CLE), University of Lausanne, Ch. des Boveresses 155, 1011 Epalinges, Switzerland; francis.derouet@unil.ch; 6Vaccine Formulation Laboratory, University of Lausanne, Ch. des Boveresses 155, 1066 Epalinges, Switzerland; christophe.barnier-quer@galvmed.org; 7GALVmed, Doherty Building, Pentlands Science Park, Bush Loan, Edinburgh EH26 0PZ, UK

**Keywords:** pandemic preparedness, technology transfer, adjuvanted vaccines, influenza, lessons learned, global health

## Abstract

Adequate global vaccine coverage during an influenza pandemic is essential to mitigate morbidity, mortality, and economic impact. Vaccine development and production needs to be sufficient to meet a vast global demand, requiring international cooperation and local vaccine production capacity, especially in resource-constrained countries. The use of adjuvants is one approach to augment the number of available vaccine doses and to overcome potential vaccine shortages. Appropriately selected adjuvant technologies can decrease the amount of vaccine antigen required per dose, may broaden or lengthen the conferred protection against disease, and may even allow protective single-dose vaccination. Here we describe a technology transfer collaboration between Switzerland and Indonesia that led to the establishment of a vaccine formulation platform in Surabaya which involved the transfer of equipment and expertise to enable research and development of adjuvanted vaccine formulations and delivery systems. This new Indonesian capability aims to facilitate local and regional access to know-how relating to adjuvanted vaccine formulations, thus promoting their application to local vaccine developers. In this review, we aim to share the “lessons learned” from this project to both support and inspire future scientific collaborations of a similar nature.

## 1. Introduction

Influenza outbreaks are rare but recurring events with potentially catastrophic consequences. In 2005, H5N1 virus spread from poultry to humans in Indonesia, with a case fatality rate of 50–70% [1]. This Highly Pathogenic Avian Influenza (HPAI) virus subtype is currently endemic in poultry in Indonesia, as well as in other countries in Asia and Africa, resulting in a sustained global threat [2]. Resource-constrained populations are generally disproportionately affected by the health impacts of pandemics. Their ability to access pandemic vaccines is limited because vaccine supplies are usually secured by high-income countries through advance-purchase agreements, as was the case during the 2009 pandemic (H1N1) [3] and currently with the COVID-19 pandemic [4]. As a shortage of vaccines is likely to occur in any pandemic situation, it is important to maximize the local preparedness and availability of a vaccine in advance for the entire population. Here we report on a multidisciplinary collaboration between Swiss and Indonesian researchers involving the development and transfer of several adjuvanted pandemic influenza vaccine technologies. The challenges and lessons learned during this scientific collaboration are described with the aim of demonstrating the feasibility of establishing a sustainable vaccine formulation platform in Indonesia, and to support access to new enabling technologies in a place where they are critically needed.

## 2. Pandemic Influenza Vaccine Development

Prophylactic vaccines are recognized as one of the most effective methods to mitigate the spread of infectious disease [5]. These benefits become most notable when the accessibility of vaccines and global vaccination rates reach sufficient levels. To achieve equitable access to vaccines international collaborations are a prerequisite. Public and private stakeholders such as the World Health Organisation (WHO) [6], Bill and Melinda Gates Foundation (BMGF) [7], Coalition for Epidemic Preparedness Innovations (CEPI) [8], and Global Alliance for Vaccine Initiative (GAVI) [9] contribute considerably to accelerate vaccine development. More recently, the COVID-19 Vaccines Global Access (COVAX) initiative has been of great importance for lower middle-income countries (LMICs) to access COVID-19 vaccines [10]. Collaboration between these various institutes is indispensable, however, lower-income countries remain mostly dependent on the political goodwill of higher-income countries for adequate supply of pandemic vaccines.

In light of the current global influenza vaccine production capacity, it is unlikely that there would be sufficient vaccine available in the first 12 months of an influenza pandemic to meet global needs. Production capacities to provide a monovalent inactivated influenza vaccine at 15 μg HA per dose, as currently used in seasonal influenza vaccines, are estimated at 4 to 6 billion doses [11,12]. A study from the 2009 swine flu pandemic showed that eight months after the first detection of the virus, less than 500 million doses of monovalent vaccine of any sort were produced [13], which would have been woefully insufficient in the case of a pandemic of higher severity. On top of this, with the suboptimal immunogenicity of pandemic influenza strains, vaccination would probably require a larger amount of antigen per vaccine dose and likely need two doses [14,15]. Consequently, there is still a need for the improvement of existing pandemic influenza vaccines. Novel egg-independent technologies are still very much in progress today, although they have started to be implemented for seasonal vaccines in certain parts of the world [12,14,16]. It is however questionable if some of these technologies could be implemented in LMICs on a large scale and within a suitable timeframe, in the case of a new rapidly spreading pandemic. Access to advanced vaccine technologies that are adapted to local capacities (such as egg-based influenza production) are therefore essential.

Ensuring adequate access to vaccines represents only one part of the multi-faceted approach of “pandemic preparedness” in developing countries. Preparedness is dependent on various parameters, such as the public healthcare system, infrastructure, economic resources, and scientific capabilities, which directly impact the development of vaccine production capacity [17,18,19]. All of these take time to establish, especially the scientific capabilities, considering that years of research generally precede the development of a vaccine. Accelerating local research and development that focuses on safe and effective vaccines is one essential component of pandemic preparedness.

The transfer of adjuvanted vaccine technology and know-how (free of intellectual property) can reduce the barriers and timelines currently associated with next-generation vaccine development in LMICs [20,21,22]. Comprehensive reports on the importance of technology transfer have been published by the United Nations [23], International Federation of Pharmaceutical Manufacturers and Associations (IFPMA) [24], and WHO [25]. There are multiple reports detailing successful influenza manufacturing technology transfers that have improved production capacities in LMICs [26,27,28,29]. However, there are only a few publications that have reported on adjuvant technology transfer, namely the transfer of oil-in-water emulsion adjuvant manufacturing technologies [30,31,32,33]. The transfer of adjuvant technologies and training of local staff, combined with the setting up of adequate facilities, would facilitate vaccine research and development in LMICs and support members of the vaccine community, such as public sector institutions, small biotechnology companies, and vaccine developers [34,35].

In this project, we focused on the technology transfer of adjuvanted influenza vaccines based on whole inactivated virion (WIV). WIV vaccines require fewer purification steps allowing better yields of production when produced by using classical egg-based technologies. A method predominantly employed by several LMIC vaccine manufacturers [36,37]. The particulate structure of WIV and its repetitive surface antigens may be beneficial for enhanced B cell activation and Th_2_ responses [38], while the presence of viral RNA has been reported to induce superior Th_1_ cellular immune responses [39,40]. Furthermore, immunological responses induced by WIV may also recognize determinants derived from conserved internal proteins and allow cross-reactivity among different subtypes [41,42]. Subsequently, WIV-based vaccines can play an important role against pandemic influenza.

## 3. Adjuvant Technologies for Pandemic Influenza Vaccines

Adjuvants can facilitate an improved immune response to pandemic influenza vaccines [43,44]. The beneficial immunostimulatory effect of adjuvants can (i) result in dose-sparing by improving antigen presentation to immune cells or achieving depot effects [45,46], (ii) enable sufficient immune responses in the elderly [47], (iii) modulate the type of immune response towards Th_1_ or Th_2_ [44], and (iv) possibly allow for the use of single-dose pandemic vaccines [48,49]. Adjuvants included in licensed pandemic influenza vaccines include aluminum salts (Al(OH)_3_ and AlPO_4_) and oil-in-water emulsions, such as MF59^®^, AS03, and AF03 (Table 1). This represents a somewhat limited choice of adjuvants for inclusion in a future influenza pandemic vaccine. Proprietary restrictions and access to suitable adjuvants are frequently considered to be a blocking point in the field of vaccine development. In addition, the know-how to properly select and formulate adjuvants with vaccine antigens is often lacking. As such, it is important to consider that the mode of action of adjuvants, the immunophenotype of the disease, and even the choice of preclinical models can play an important role in distinguishing the safety and efficacy of an antigen–adjuvant combination.

The described technology transfer collaboration encompassed both the physical establishment of an adjuvant vaccine formulation platform, along with the research and development of adjuvanted single-dose pandemic influenza vaccines. The adjuvants selected were SWE (a squalene-in-water emulsion) and LQ (a liposome-based adjuvant containing the QS21 saponin). SWE is similar in composition to MF59™, which has an extensive safety record in humans in the context of seasonal and pandemic influenza vaccines. Dose sparing capacity has been demonstrated with influenza vaccines for both MF59™ in humans [51,52] and SWE in pre-clinical studies [45]. Moreover, SWE is manufactured at GMP grade (as Sepivac SWE™) and is openly accessible to the vaccine community. The LQ adjuvant is based on a mature liposome technology consisting of DOPC and cholesterol, combined with QS21. QS21 is an immunostimulatory molecule that is included in the AS01 adjuvanted Mosquirix™ and Shingrix vaccines [53]. LQ was evaluated for its potential to trigger T cell responses and was of interest for this technology transfer due to its ease of preparation at lab-scale, whilst also being readily upscaled.

## 4. The Indonesian Scientific Infrastructure; Challenges and Needs for Vaccine Development

Historically, HPAI H5N1 virus has been circulating in Indonesia and has become endemic in poultry and other animals [2,54]. Direct animal-to-human transmission of H5N1 has resulted in numerous human deaths (455 deaths out of 862 confirmed cases over the period 2003–2020) [55]. With the fourth largest human population globally, spread over a wide and fragmented geographic area, Indonesia is a particularly vulnerable location for a H5N1 pandemic outbreak, should this subtype attain the capability to become transmissible between humans. Subsequently, pandemic preparedness through local influenza vaccine development in Indonesia is a particularly important tool.

The resources and expertise for working with modern vaccines are often not readily accessible in the countries that are most affected by pandemic threats [56], something that has been particularly highlighted by the COVID-19 pandemic [57]. The combination of technology transfer and local support from governments for scientific development could reduce this gap. The presence of vaccine manufacturers, research, and a strong public health policy in Indonesia are very encouraging factors which could undoubtedly support the integration of innovative vaccine technologies to strengthen the pandemic preparedness of the local population. Moreover, the Indonesian Ministry of Health has implemented a National Committee on adverse events following immunization (AEFI) [58], similar to the Vaccine Adverse Event Reporting System (VAERS) for U.S. licensed vaccines [59], or the European Vaccination Information Portal [60]. Of note, the importance of strong growth and independence of the local Indonesian vaccine industry was recently highlighted by the Indonesian Minister of Research and Technology [61].

Since 2009, Indonesia has made a considerable shift from the import of seasonal influenza vaccines to domestic vaccine production through cooperation between PT Bio Farma, Indonesia’s Ministry of Health, and several other partners such as Kobe University in Japan [62]. In doing so, Indonesia embarked on a strategy to increase pandemic preparedness, with the local manufacture of seasonal and pandemic influenza vaccines [63]. Current capabilities include the fill and finish of bulk seasonal influenza vaccine (Flubio^®^) and development of pandemic vaccines for which PT Bio Farma relies on the robust method of egg-based production to ensure rapid licensure of the vaccine in case of a pandemic outbreak [64,65]. Such a vaccine infrastructure may benefit from the use of adjuvants to enable pandemic vaccine dose-sparing and durable immune responses. In this respect, the limited access to adjuvants and formulation expertise represents a potential obstacle for pandemic preparedness in Indonesia.

The prevailing need for adjuvant know-how was reinforced in March 2017 during the “Swiss-Indonesian Vaccine Formulation” symposium in Surabaya. It was attended by a range of experts from academia, industry, international organizations, biotechs, higher education, public health representatives, and diplomatic staff. Students of undergraduate, magister/masters, or doctoral programs represented 50%, and researchers made up 27% of the participants. Both fields of veterinary (37%) and human (30%) vaccines were represented. All participants were surveyed to collect information on the need for adjuvants and vaccine formulation expertise in Indonesia. Interestingly, most participants (62%) said they had never worked with adjuvants before. Analysis of the survey answers highlighted the current need for increased adjuvant expertise in Indonesia; 81% of participants indicated that they “would” or “would possibly” need access to adjuvants in the future (Figure 1a). When asked which adjuvants would be most needed, oil-in-water emulsions, liposomes/nanoparticles, and aluminum salts represented 48% of their needs, while 37% answered they would need advice on the selection of the most appropriate adjuvant (Figure 1b). With most respondents stating that they saw a current need for vaccine formulation expertise and support. This strongly validated and encouraged the project’s aim of establishing an Indonesian vaccine formulation platform.

## 5. A Swiss-Indonesian Collaboration

For this project, a research consortium was established within the scope of the Swiss Programme for Research on Global Issues for Development (r4d). This is a Swiss funding scheme which supports high-quality research projects that aim to provide solutions to global problems with a focus on collaborations between Swiss and LMIC organizations. One of the Swiss partners was the Vaccine Formulation Laboratory (VFL) at the University of Lausanne, today known as Vaccine Formulation Institute (VFI), an independent not-for-profit company. Briefly, the VFI supports the vaccine community with gaining wider access to adjuvant technologies and vaccine formulation know-how [35]. A second Swiss partner, the unit of Biopharmaceutical Sciences at the University of Geneva (UNIGE), is dedicated to the formulation, characterization, and testing of advanced drug and vaccine carrier systems. Over the last ten years, they have developed expertise covering protein drugs, DNA vaccines, novel adjuvants, and nanomedicines. The Indonesian partner was the Professor Nidom Foundation (PNF) with support from Airlangga University. The PNF is an Indonesian research and education organization that has extensive experience in the surveillance and characterization of influenza viruses. Research activities at the PNF include establishing recombinant influenza seed virus for improved vaccine virus yield and developing alternative cell-culture-derived virus propagation tools, using zebrafish primary cell cultures. An additional Indonesian collaborator of this project was PT Bio Farma, the largest vaccine manufacturer in Indonesia, who provided their expertise and support on vaccine production. PT Bio Farma is a state-owned vaccine producer and the only producer of human vaccines and anti-sera in Indonesia. In 2010–2015, PT Bio Farma was a technology transfer recipient through a project funded by the United States Department of Health and Human Services’ Biomedical Advanced Research and Development Authority (BARDA), in the context of a WHO international collaboration on technology transfer. Within this project the VFL transferred the know-how to PT Bio Farma for manufacturing an oil-in-water emulsion [32].

## 6. The r4d Swiss-Indonesian Technology Transfer Project

### 6.1. Project Outcomes

An incentive to drive technological advancement is a basic need for a technology transfer to occur and many factors can influence the incentive, such as threat-related geographical position, and need and demand [23]. The need and demand for adjuvant technology transfer in Indonesia was confirmed by survey responses from the Swiss-Indonesian symposium, as described previously. As a result, the symposium generated local interest from both human and veterinary vaccine researchers, which subsequently nurtured the ecosystem and research network required to sustain the transferred technologies.

Further growth of this research network is expected through the established vaccine formulation platform in Surabaya, consisting of a new laboratory that is adequately equipped for adjuvanted vaccine formulation research and development. It is important to note that this new platform builds on an existing scientific infrastructure and suitable level of skill [66,67]. Consequently, ongoing collaborations and partnerships with neighboring laboratories and industry partners were strengthened during the course of this project. As an example, PT Bio Farma provided in-kind support to PNF staff in the preparation of the Indonesian H5N1 WIV vaccine candidate. Demonstrating how existing scientific capacity and partnerships supported the research component of this project. PT Bio Farma was earmarked in this project as a potential industrialization partner, while the PNF was identified to manage small-scale academic collaborations to support the further development and dissemination of Indonesian vaccine expertise.

The adjuvant development activities were conducted jointly by both Swiss partners (VFL and UNIGE), while vaccine preparation was performed by the Indonesian partner (PNF). To this extent, SWE and LQ adjuvanted WIV influenza vaccine formulations were identified as potential candidates for an adjuvanted single-dose immunization strategy. SWE was available on an open access basis and locally from PT Bio Farma. The equipment and materials to manufacture liposomes at lab scale were acquired locally or transferred through the project.

In addition, suitable polymers and optimal parameters for microparticle manufacturing were investigated for the encapsulation of WIV. The particles were evaluated as a preliminary research activity for a single-dose delayed release strategy.

The adjuvanted vaccine strategies were developed in Switzerland with a model WIV H5N1 antigen (A/turkey/Turkey/1/2005 (NIBRG-23)), and for the vaccine candidate to be more applicable to LMIC vaccine development, the Indonesian partners produced their own WIV H5N1 antigen (A/Indonesia/D4/2007). Quality control of the vaccine candidate was performed locally in Indonesia and also in Switzerland. Following the development of adjuvanted and delayed-release approaches, technology transfer to the PNF included extensive training both onsite and in Switzerland. The transfer was concluded with the evaluation of in vivo immune responses to the adjuvanted WIV influenza vaccine candidates. Cellular and humoral responses in mice following a single-dose of LQ adjuvanted WIV demonstrated comparable immunogenicity as induced by a prime and boost of the antigen alone. These results are further discussed in an upcoming publication.

### 6.2. Project Challenges

#### 6.2.1. Multicultural Collaboration

The success of international scientific projects is often measured by their scientific achievements, but such success is founded on effective communication and good working relationships. In this respect, face-to-face meetings that combined project management topics and scientific discussions were found to be essential, in addition to regular teleconference meetings, using suitable equipment and reliable internet connections. In further support of this, a two-day Swiss-Indonesian Vaccine Formulation symposium was organized giving a platform to both international speakers and local researchers, including a range of experts from academia and industry from the vaccine development field. At each opportunity, theoretical presentations were demonstrated with a practical workshop on adjuvant formulation. The added value of social activities should not be underestimated for improving communication, awareness of partner organization dynamics, and understanding of cultural differences.

#### 6.2.2. Knowledge Transfer

It is well-known that scientific technology transfer sometimes results in the installation of (usually expensive) equipment with inadequate investment in transferring know-how to the recipients [20]. To avoid this pitfall, this project focused on repeated visits, e.g., an initial theoretical introduction and practical training on the technology in Switzerland, followed by a repetition in Indonesia using the local equipment and facilities. Not only is this aspect important for the sustainability of the technology transfer, but, for vaccine development, it is essential to confirm the reproducibility of the data generated by the new platform. By repeating the visits, ongoing corrections and adaptions of the technology to the conditions and circumstances in Indonesia were possible, as well as a comprehensive translation of operating procedures to different laboratory settings and a different language.

#### 6.2.3. Constraints in Local Availability and Transfer of Scientific Supplies and Equipment

The regional unavailability of several laboratory supplies and specialized equipment raised significant barriers to the progress of the project. The import of laboratory supplies was necessary in some cases, as the quality control of locally produced laboratory materials was lacking. The project also encountered several delays due to limited accessibility of specific reagents. This was sometimes due to a lack of regional distributors or, for example, the Indonesian import regulations for certain scientific materials were not yet clearly described or there were differences in how regulations were written and applied between local, regional, or national levels.

The exchange of materials between project partners was another major challenge. Each country has specific rules that may complicate material transfer and therefore access to biological samples or virus samples [68]. For example, it was important to note that Indonesia had previously introduced the idea of protection of viral samples, and initiated a change in global mindset towards a collective virus- and benefit-sharing system [69]. The protection of such resources has been described in the “Nagoya Protocol on Access to Genetic Resources and the Fair and Equitable Sharing of Benefits Arising from their Utilization (ABS) to the Convention on Biological Diversity” [70].

## 7. Lessons Learned

The lessons learned in respect to this project are summarized in Table 2.

The scientific symposium held at the start of the project sparked the interest of the local research community. We learned that this was beneficial in order to establish fruitful partnerships and collaborations, subsequently sustaining later research activities. When writing the project plans, the ability of the scientific infrastructure of the transfer recipient to sustain research activities should be carefully outlined and then confirmed. This can be a major limiting factor for research and development, especially if the technology transfer includes novel or advanced technologies. Even with highly educated and motivated people and adequate facilities, technology transfer projects to LMICs may be strongly impeded by national import/export customs regulations. This project suffered major issues due to an inability to import critical materials into Indonesia. A lesson learned in this project was to preferably establish import routes before the start of the project, possibly through contacting relevant specialist local or regional companies who provide their own logistics and import networks. It is worth noting that several global courier companies also encountered difficulties in successfully importing required items.

Complex projects which are both multi-cultural and multi-disciplinary should carefully consider the timelines set for individual activities, and these should be discussed regularly and be revised as necessary. A lesson learned to reduce the risk of delays was the importance of the regular and thorough evaluation of results, ongoing activities, and longer-term planning. In this respect, it was helpful to assign the responsibility to a dedicated Project Lead who was supported by Project Managers at all partner sites. The Project Lead should be “hands-on” and possess a mix of administrative and scientific expertise. They should strive to provide an “enabling” environment to ensure that the Project Managers and project members are able to function efficiently and in a coordinated fashion. For this it is important that all relevant parties are properly involved in decision-making, troubleshooting, and risk management. Ideally, decision-making should be by consensus, with no single person in charge, so as to harbor trust and transparency and ensure the engagement of all persons involved.

In terms of communication, another essential lesson learned was that regular project teleconferences were continued until the very end of the project, even if there were no scientific results to report for a given period of time. Recurring meetings are crucial in maintaining an ongoing high level of trust, knowledge transfer, and collaboration between project partners. Opportunities should be sought to combine regular project meetings with training on the transferred technologies, and face-to-face meetings should be held at all locations as much as possible. A notable lesson learned was the value of repetition of training in an ongoing fashion (both practical and theoretical) at both Swiss and Indonesian partner facilities. The selection of both equipment and materials should ideally be based on common accessibility. If this is not possible, scientific protocols should anticipate the use of different materials and equipment, and comparative studies should be planned accordingly.

The success of this project also depended largely on the funding body who facilitated the required changes in milestones and/or timelines as the project progressed. Transparency between project partners and the funders is therefore essential.

## 8. Conclusions

Due to the ever-present risk of an influenza pandemic outbreak, access to globally produced vaccines or time to market approval of locally produced vaccines in LMICs needs to be reduced. Although equitable access to vaccines produced by other countries is of paramount importance, enabling local production of vaccines in combination with appropriate adjuvant technologies would help meet the global demand much faster. In an ideal situation, sufficient preparedness for a pandemic influenza outbreak will allow a combination of local and global vaccine development responses to rapidly mitigate and stop the spread before the outbreak moves from epidemic to pandemic. In this project, an adjuvant technology transfer was performed from Switzerland to Indonesia, including both the equipment and formulation expertise. The potential of the selected adjuvant technologies in combination with a locally prepared WIV antigen was evaluated as a pandemic influenza vaccine candidate. In addition, the successful establishment of an Indonesian vaccine formulation platform with properly trained local scientists was an important milestone for Indonesia and the surrounding regions to access adjuvants and the associated knowledge on how to use them. In preparation for the next pandemic, Indonesia now has an additional tool for the development of adjuvanted vaccine formulations.

## Figures and Tables

**Figure 1 vaccines-09-00461-f001:**
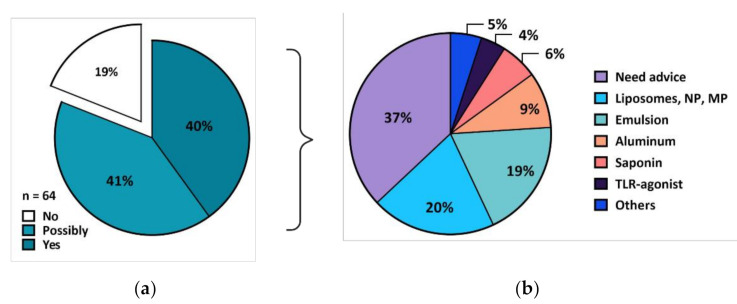
Survey responses of 64 participants. (**a**) When asked if they would need an adjuvant to support their work, 81% of participants “would” or “would possibly” need access to adjuvants; (**b**) various types of adjuvant were of interest with liposomes, nanoparticles (NP), and microparticles (MP) making up 20%, mulsion-based adjuvants were of interest to 19% of participants. Notably, 37% of participants indicated that advice was needed.

**Table 1 vaccines-09-00461-t001:** Adjuvants included in licensed pandemic influenza vaccines summarized from References [12,44,50].

Name	Category	Components	Product Name(s)
Alum ^1^	Mineral salt	Al(OH)_3_, AlPO_4_, AlPO_4_ gel	Daronrix ^2^, Orniflu, Panflu, Emerflu ^2^, Fluval-H5N1
AF03	Oil-in-water emulsions	Squalene; polyoxyethylene cetostearyl ether; mannitol; sorbitan oleate	Humenza ^2^
AS03	Oil-in-water emulsions	Squalene; α-tocopherol; polysorbate 80	Arepanrix ^2^, Pandemrix ^2^, Prepandrix ^2^
MF59^® 1^	Oil-in-water emulsions	Squalene; polysorbate 80; sorbitan trioleate	Aflunov, Focetria ^2^

^1^ Also licensed for seasonal influenza vaccines, ^2^ No longer in use/marketing authorization withdrawn from use in the EU.

**Table 2 vaccines-09-00461-t002:** An overview of key lessons learned that can be applicable to technology transfer projects in lower middle-income countries within the scope of adjuvanted vaccine development.

Topic	Key Lesson Learned
Feasibility	Ensure there is a need and an ecosystem to sustain the transferred technologies
Planning	Consider local scientific infrastructure, and preferentially establish reliable import/export routes before the start of the project
Review	Evaluate scientific results periodically for ad hoc troubleshooting
Decision making	Ensure an enabling environment that promotes collaborative and decisive decision making in a pre-agreed manner
Communication	Hold regular meetings for the entire duration of the project
Communication	Meet as frequently as possible at face-to-face meetings at both (or all) partner locations
Technology	Perform repetitive training at both locations with the same people and types of equipment
Technology	Select materials that are locally accessible at the recipient site or anticipate using different materials depending on availability
Funding	Cultivate good working relationships with any funding bodies to ensure their advice and support when difficulties are inevitably encountered

## Data Availability

MDPI Research Data Policies.

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
