# Peer review of "Better Pandemic Influenza Preparedness through Adjuvant Technology Transfer: Challenges and Lessons Learned"

_vaccines, 2021, doi:10.3390/vaccines9050461_

Round 1

Reviewer 1 Report

The manuscript entitled “Better Pandemic Influenza Preparedness through Adjuvant Technology Transfer; Challenges and Lessons Learned” is not suitable for publication in “Vaccines”. Although this manuscript is well-written as a commentary, it is totally informative. They describe a technology transfer collaboration between Switzerland and Indonesia that led to the establishment of a vaccine formulation platform in Surabaya which involved the transfer of equipment and expertise to enable research and development of adjuvanted vaccine formulations and delivery systems, whereas they never mentioned how they will do this transfer or how Surabaya will produce and protect this technology. They chose Indonesia but there a lot of lower middle-income countries in Asia and Africa, it is not clear that this transfer can also reach these countries, too. It is also not clear why adjuvants such as SWE (a squalene-in-water emulsion), and LQ (a liposome-based adjuvant containing the QS21 saponin) were selected. The amount of them and superiority of them over the others are not well explained, either. It is mentioned that an adjuvant technology transfer was performed from Switzerland to Indonesia including both the equipment and formulation expertise. However, there are not any quantitative results about this transfer. Whole manuscript looks so theoretical and subjective without any quantitative data. Therefore, I recommend rejection.

Here are the some points below:

  • It shouldn’t be a space between “%” and numbers; such as “20 %”.
  • The written style of Reference 54 should be corrected.
  • They used “et al.” in some References but they did not use some of them (more than 3 or 4 authors). It must be standard.

Author Response

Dear Reviewer,

We appreciate the time you have invested to provide your comments on our manuscript. Please see the attachment for our responses.

Sincerely, on behalf of all authors,

Céline Lemoine

Reviewer 2 Report

This is a "news article"  about a subject that is not new and for which no result-was a suitable adjuvant vaccine produced and shown to be beneficial? Other initiatives in Indonesia have been published before and also in South America

I think the paper needs and editorial decision about how important a work in progress without results is for a research journal especially given the authors have web site with the project on it.

I do not consider the tabulated views  of a mixed undefined  conference audience is of any value (outside of the conference)

Author Response

(The authors gave the same response as above.)

Reviewer 3 Report

The authors summarize a collaboration between two institutions to strengthen influenza vaccine preparedness. This should be classified as review article. Although of interest to the vaccine community to showcase how collaborative efforts can bolster preparedness, I have few minor suggestions.

  • A table of adjuvants in use for influenza vaccination could be helpful.
  • A note on how the AS03 adjuvanted pandemic infleunza vaccine (Pandemrix) was associated with narcolepsy cases could be included to show that adjuvanted vaccines need to undergo extensive testing before deployment
  • A vaccine adverse event system should also be discussed.

Author Response

(The authors gave the same response as above.)

Round 2

Reviewer 1 Report

I accepted.

Reviewer 2 Report

None